# Study of Effector CD8+ T Cell Interactions with Cortical Neurons in Response to Inflammation in Mouse Brain Slices and Neuronal Cultures

**DOI:** 10.3390/ijms24043166

**Published:** 2023-02-06

**Authors:** Ching-Hsin Lin, Anja Scheller, Yang Liu, Elmar Krause, Hsin-Fang Chang

**Affiliations:** 1Cellular Neurophysiology, Center for Integrative Physiology and Molecular Medicine (CIPMM), Saarland University, 66421 Homburg, Germany; 2Molecular Physiology, Center for Integrative Physiology Molecular Medicine (CIPMM), Saarland University, 66421 Homburg, Germany; 3Department of Neurology, Saarland University, 66421 Homburg, Germany

**Keywords:** cytotoxic CD8+ T cells, neuron, cortical culture, acute brain slice, T cell–neuron co-culture, T cell migration, cytokine, inflammation, neurodegeneration, autoimmune disease

## Abstract

Cytotoxic CD8+ T cells contribute to neuronal damage in inflammatory and degenerative CNS disorders, such as multiple sclerosis (MS). The mechanism of cortical damage associated with CD8+ T cells is not well understood. We developed in vitro cell culture and ex vivo brain slice co-culture models of brain inflammation to study CD8+ T cell–neuron interactions. To induce inflammation, we applied T cell conditioned media, which contains a variety of cytokines, during CD8+ T cell polyclonal activation. Release of IFNγ and TNFα from co-cultures was verified by ELISA, confirming an inflammatory response. We also visualized the physical interactions between CD8+ T cells and cortical neurons using live-cell confocal imaging. The imaging revealed that T cells reduced their migration velocity and changed their migratory patterns under inflammatory conditions. CD8+ T cells increased their dwell time at neuronal soma and dendrites in response to added cytokines. These changes were seen in both the in vitro and ex vivo models. The results confirm that these in vitro and ex vivo models provide promising platforms for the study of the molecular details of neuron–immune cell interactions under inflammatory conditions, which allow high-resolution live microscopy and are readily amenable to experimental manipulation.

## 1. Introduction

Cytotoxic CD8+ T cells play an essential role in adaptive immunity. They recognize targets via the major histocompatibility complex (MHC) class I. Recognition triggers a cascade of T cell activation and signaling, resulting in the release of effector molecules, such as lytic substances and cytokines, to destroy target cells or to enhance the inflammatory response. Neurons and other brain cells express MHC I molecules and are susceptible to targeting by CD8+ T cells [1,2]. Neural antigen-reactive cytotoxic CD8+ T cells contribute to neuronal dysfunction and degeneration in a variety of inflammatory central nervous system (CNS) disorders, such as multiple sclerosis (MS), neuromyelitis optica, and acute disseminated encephalomyelitis [3]. MS is the most common of these diseases. Both myelinated white matter and gray matter [4,5,6] are affected by this autoimmune attack. The mechanisms of gray matter inflammation and degeneration in MS in the context of T cell autoimmune attack are not well understood. Experimental autoimmune encephalomyelitis (EAE) is a commonly used animal model for MS. In EAE, neural damage is caused by infiltrating autoreactive T cells, which react to neural antigens, such as myelin oligodendrocyte glycoprotein (MOG) and myelin basic protein (MBP), and attack the myelin sheath of neurons, resulting in CNS inflammation, demyelination, and neurodegeneration [7,8,9].

Both CD4+ and CD8+ T cells are found at lesion sites in MS patients and EAE animals [10], but CD8+ T cells are the predominant T cell population in human MS lesions [11]. Their role in autoimmune pathologic processes remains unclear. Cytotoxic CD8+ T cells are considered important effector cells that contribute to neuronal damage [12,13,14], but they have been reported to be neuroprotective and to limit neuronal damage in EAE [15,16,17,18]. This suggests that CD8+ T cells may play a role in regulating the balance between neuroprotection and neural damage through their interaction with brain cells. Furthermore, the brain possesses defense mechanisms that support neuronal plasticity and protect against inflammation and neuronal damage. This complex defense mechanism involves glia cells and neurons that actively remove infiltrating immune cells or minimize neuronal damage by inducing synaptic loss [19,20,21,22,23]. Regulatory T cells play an essential role in suppressing inflammation. Neurons are reported to be highly immune regulatory. They induce proliferation of activated CD4+ T cells and convert these encephalitogenic T cells into regulatory cells via interactions with T cells that inhibit EAE [24]. The use of EAE models has contributed to our understanding of the pathology of MS, but much remains to be learned about how brain cells interact with infiltrating T cells in the context of neuroprotection and neurodestruction. 

To study how neural antigen-reactive cytotoxic CD8+ T cells interact with brain cells in inflammation, we developed in vitro neuronal cell culture and ex vivo acute brain slice co-culture models with activated T cells, allowing effector T cells to interact with neuronal cells. To visualize CD8+ T cells and neurons, we used CD8-cre-τGFP mice and Thy1.2-HcRed transgenic mice whose cortical neurons express the far-red fluorescent protein HcRed1. We used different effector T cell preparations adapted to in vitro and ex vivo co-culture studies. In addition, we generated inflammatory conditions in these two models by supplying T cell conditioned medium that contained a variety of cytokines produced by CD8+ T cells during their polyclonal activation. We verified the inflamed condition in both co-culture models. Changes in T cell migration were observed by live-cell confocal imaging, and an increase in two key proinflammatory cytokines, interferon gamma (IFNγ) and TNF alpha (TNFα), was confirmed by ELISA assay. Our in vitro and ex vivo models provide an easily manipulated platform for studying the molecular details of neuron–immune cell interaction under inflammatory conditions. Combining these two models with high-resolution imaging techniques can potentially uncover molecular details of neuropathological processes or neuroprotective mechanisms that are difficult to extract from in vivo studies due to limited visualization access and the complexity of cell populations. 

## 2. Results

### 2.1. Cortical Neuronal Culture and Acute Brain Slice Preparation for In Vitro and Ex Vivo Studies

To study T cell and neuronal cell interactions at the single-cell level and in a pathological brain environment, we first established a cortical neuronal cell culture and generated live brain slices to preserve a 3D brain structure with native brain cell composition. For neuronal cell culture, we isolated brain neocortices from postnatal day 0 to day 1 (P0–1) pups. Single brain cells were isolated from neocortices through multiple preparation steps and were finally seeded on collagen I-coated glass coverslips, allowing the neuronal cells to adhere and grow (Figure 1A). As neurons and glia cells are susceptible to environmental changes, we developed another preparation consisting of acute brain slices, providing brain cells with a more native environment. We isolated whole brains from 2–3-week-old pups and performed brain coronal sectioning using a vibratome. Isolated brains were kept in ice-cold high sucrose (230 mM) buffer during sectioning to minimize brain damage. We sectioned the brains into 350 µm thick slices and placed them in a recovery bath containing artificial cerebrospinal fluid (ACSF) at 35 °C. After a minimum 2 h recovery period, the damaged tissue at the surface of the cut was slightly dissociated, resulting in thinner sections (Figure 1B). We examined the integrity and neuronal viability of the slices after recovery by observing the morphology of neurons in neocortices from Thy1.2-HcRed transgenic mice with a C57BL/6N background strain in which cortical neurons express cytosolic far-red fluorescent protein HcRed1. The red cortical neurons were intact, and their layered structure in deeper areas of the slices was well preserved (Appendix A). To assess brain cell composition in the in vitro culture and ex vivo brain neocortex, we analyzed three major brain cell types, neurons, astrocytes, and microglia, by staining them with cell-type-specific marker anti-NeuN, anti-GFAP, and anti-IBA1 antibodies, respectively (Figure 1C). In cortical neuronal culture, approximately one-third of the population was neurons (34.6 ± 10.1%), and the rest consisted mainly of astrocytes (62.3 ± 10.5%). Microglia made up a minor fraction of the culture (3.2 ± 0.4%). In the acute brain slices, we analyzed brain cells at the region of the motor cortex and sensory cortex, where 71.7 ± 5.3% were neurons, 12.7 ± 2.8% were astrocytes, and 5.0 ± 0.2% were microglia (Figure 1D).

### 2.2. Effector T Cell Generation for Co-Cultures

To generate activated effector T cells and to study how they interact with inflamed neuronal networks, we positively isolated naive CD8+ T cells from the splenocytes of CD8cre-τGFP transgenic mice in which tubulin-GFP is specifically expressed in the CD8+ T cell population. The isolated cells were then polyclonally activated with anti-CD3/anti-CD28 antibody-bound dynabeads supplemented with 50 U/mL mouse IL-2 recombinant protein for 2 days. Activated blast cells formed after 2 days. We doubled the supplemented IL-2 amount to 100 U/mL to allow the effector cells to proliferate. After 3–4 days of activation from the naive stage, these T cells were introduced to neuronal cells in vitro (Figure 2A). Notably, due to a lack of oligodendrocytes in our neuronal culture, no myelin formed in the neuronal network. Thus, we used dynabeads to generate effector T cells that did not prime to myelin-associated antigens for the in vitro study.

For the ex vivo study, we primed the T cells with the CNS antigen, MOG peptide, via antigen-presenting cells in the splenocytes. This priming led the T cells to specifically recognize myelin on neurons. Naive T cells embedded in the splenocytes were primed with 75 µg/mL MOG_35-55_ peptide supplemented with 50 U/mL mouse IL-2 and a small amount of suspended anti-CD3 and anti-CD28 antibodies to assist T cell activation. After expanding the MOG-specific T cells, we performed Ficoll density gradient centrifugation to purify the lymphocyte population from the splenocyte pool. Effector T cells were used to co-culture with the brain slices for functional assay after the second cycle of stimulation (second restimulation) (Figure 2B). We analyzed the MOG-specific lymphocyte population isolated from the WT mice in the culture by labeling anti-CD4 and anti-CD8 antibodies on these cells. In the second cycle of stimulation, on day 4, 61.9 ± 2.9% of these cultures were composed of CD8+ cells, 25.5 ± 2.0% of CD4+ cells, and 12.6 ± 2.7% of CD8-CD4− cells. The double-negative cells were most likely B lymphocytes (Figure 2C). Notably, the MOG-specific T cells generated from the WT mice and CD8cre-τGFP transgenic mice yielded lymphocyte populations of similar composition in the culture, using the protocol described in Figure 2B. We applied these restimulated MOG-specific lymphocytes to the brain slices for the ex vivo study. Of note, the protocols for preparing cortical neuronal culture and acute brain slices used for measuring neuronal activity and neurogeneration [25,26,27,28,29], as well as effector T cells [30,31] and their co-culture with neurons to investigate T cell effector function and interplay with neurons, have been reported before [32,33,34]. Here, we describe cortical cultures adapted to our experimental needs, followed by effector T cell co-cultures.

The heterogeneity of MOG-specific T cells offers the advantage of studying neurological pathologies induced by neural antigen-reactive T cells that coherently communicate with each other during inflammation. As we used isolated cells from CD8cre-τGFP mice, we visualized only the CD8+ T cell population. Thus, we created a rather comprehensive immune environment mimicking autoimmune neuropathology and focused on CD8+ T cells. We utilized different protocols to generate effector T cells to co-culture with neurons in both in vitro and ex vivo brain slices to study T cell–neuron interaction. 

### 2.3. CD8+ Effector T Cells Interact with Neurons in Inflamed 2D and 3D Neuronal Networks

To visualize how effector T cells interact with neurons, we co-cultured bead-activated CD8cre-τGFP T cells (green) with cortical neuronal cells and observed T cell migration using live confocal imaging. Neurons were labeled with the mitochondrial marker cox8a-mscarlet (red) by infection with lentivirus under a neuron-specific synapsin promotor. T cells migrated on the neuronal layer with variable patterns and velocity and with random direction. We observed migrating CD8+ T cells that changed direction when encountering neurons. Most of the T cells became stationary upon contact with neurons, including somatas and neurites. In Figure 3, a green CD8+ T cell moved rapidly from a neuron-free area until contacting a dendrite (5:36 min) and a second dendrite (11:12 min), resulting in a slower migration speed. This T cell moved further and contacted a third dendrite (20:32 min), where it remained for about 20 min (until 39:40 min). An early step in immune synapse formation is translocation of the microtubule-organizing center (MTOC) toward the contact between the immune cell and target cell, which is initiated by T cell receptor signaling [35,36,37]. The interaction between CD8+ T cells and neurons can be recognized by the appearance of the tubulin-GFP-marked MTOC of the T cells at the encountered neurites. Occasionally, T cells migrate along dendrites before or after pausing (Figure 3A; Appendix A). 

Normally, the brain is protected from infiltration, and T cells are unable to reach this immune-privileged region [38]. However, significant contact between immune cells and brain cells can occur during pathogenesis accompanied by inflammation when vascular permeability in the CNS increases. We produced the inflamed condition by applying T cell conditioned medium collected from CD8+ T cell culture during the first 48 h of polyclonal activation (Figure 2A). This conditioned medium contained a variety of cytokines, including proinflammatory cytokines IFNγ (about 738 pg/mL) and TNFα (about 176 pg/mL), which we have referred to as the cytokine cocktail. We measured the release of these two cytokines from three independent cultures. The amount of these two cytokines appeared to be quite stable in each culture, as the T cell activation protocol is tightly controlled. We applied this cocktail to the co-culture and examined how the T cells migrated on the neuronal layers in the presence of these cytokines (Figure 3B).

We first added T cells to the neuronal culture, allowed them to adhere and migrate on the neuronal layer and recorded their migration for 45 min (before). Then, we applied the cytokine cocktail to the co-culture and recorded for another 45 min (after). We analyzed various migration parameters, including velocity, straightness, track length, and displacement. Upon cytokine application, each of the analyzed parameters decreased significantly. The values were: velocity (before: 4.48 ± 0.18 µm/min; after: 3.87 ± 0.12 µm/min), migration straightness (before: 0.47 ± 0.02; after: 0.39 ± 0.02), track length (before: 68.52 ± 4.38 µm; after: 50.52 ± 2.79 µm), and displacement (before: 28.35 ± 2.75; after: 16.65 ± 1.53) (Figure 3C, mean ± SEM; N = 3, *n* = 143 for before and *n* = 148 for after). The slower migration was not due to technical artifacts of prolonged live-cell recording since, under the same co-culture conditions, the T cells showed unchanged migration patterns and velocity in the absence of cytokine application (Appendix A; N = 2, *n* = 58 for before and *n* = 69 for after). These results suggest that T cells become stationary and decrease their migration velocity when in contact with soluble cytokines, which mimic the inflamed condition in neuronal cultures. 

In many neurodegenerative disorders, neuroinflammation is associated with infiltration of T cells. Whether inflammation leads to a change in T cell migration is not clear. To address this point, we pre-incubated neuronal cells with the proinflammatory cytokine cocktail for 12 h or 24 h to create an inflamed neuronal network and applied T cells to record their migration (Figure 3D). CD8+ T cells migrated much more slowly (without: 7.48 ± 0.2 µm/min; pre-incubate 12 h: 5.28 ± 0.13 µm/min) and had shorter tracks (without: 109.40 ± 4.92; pre-incubate 12 h: 102.75 ± 3.85) that were less straight (without: 0.5 ± 0.02; pre-incubate 12 h: 0.37 ± 0.02) and showed less displacement (without: 51.81 ± 3.07; pre-incubate 12 h: 36.59 ± 2.10) on the 12 h cytokine-treated culture. After 24 h of treatment with the cytokines, the changes in migration parameters reverted and approached control levels (migration velocity (5.74 ± 0.2 µm/min), migration straightness (0.44 ± 0.02), track length (109.32 ± 6.0 µm), and displacement (43.60 ± 3.45)). Despite a significant increase in T cell migration speed in 24 h cytokine pre-incubated cultures compared to 12 h pretreatment, the T cells were less mobile on neuronal cells pretreated with the cytokine cocktail (Figure 3E, mean ± SEM; N = 2; *n* = 163 for the without group, *n* = 193 for the 12 h group, and *n* = 193 for the 24 h group). These data demonstrate that CD8+ T cells reduce their migration velocity and alter their migration pattern when in contact with cytokine-pretreated neuronal cells. 

Next, we observed T cell migration in ex vivo brain slices. We carried out these experiments in cultured brain slices, as described above. The neurons expressed the far-red fluorescent protein HcRed1. We applied MOG-specific CD8cre-τGFP T cells (green) on the brain slices and observed T cell migration using confocal microscopy. The best quality of both T cells and acute brain slices was achieved using ACSF as the buffering medium in the co-cultures. As with the in vitro culture, the CD8+ T cells exhibited variable migration velocities and patterns in the cortical slices. We observed changes in migration when the T cells encountered neurons (red). In Figure 3F, for example, live imaging captured a CD8+ T cell that approached a cortical neuron (time 2:20 min) and dwelled at its soma (until 9:20 min) before moving on (until 12:36 min) (Figure 3F, closed white arrows). Another T cell encountered a neuron at 36:24 min and stopped but quickly moved further (37:20 min and 38:16 min) (Figure 3F, opened white arrows; Appendix A). We then analyzed T cell migration after cytokine cocktail application to mimic the inflamed condition (Figure 3G). The T cells reduced their velocity (before: 5.5 ± 0.28 µm/min; after: 3.34 ± 0.18 µm/min) and straightness (before: 0.38 ± 0.03; after: 0.39 ± 0.03) after the cytokines were added, whereas track length (before: 67.52 ± 6.17 µm; after: 73.52 ± 5.14 µm) and displacement (before: 25.65 ± 3.47; after: 20.14 ± 2.53) were unchanged (Figure 3H). 

We examined the effects of immediate (Figure 3B,C) and prolonged treatment with cytokines (12 and 24 h; Figure 3D,E) to mimic inflammation by applying the cytokine cocktail to the neuronal culture in vitro. CD8+ T cells exhibited reduced migration velocity and altered migration patterns under all these conditions compared to the untreated groups. Moreover, the changes in T cell migration were similar in the inflamed cortical brain slices ex vivo (Figure 3G,H). These results indicate that the T cells responded similarly to the inflammatory conditions in both models.

### 2.4. Proinflammatory Cytokine Release in Cortical Neuronal Culture and Brain Slice during T Cell Co-Culture

We next analyzed the release of two key proinflammatory cytokines, interferon gamma (IFNγ) and TNF alpha (TNFα), over time in both the in vitro and ex vivo co-cultured models to verify the inflamed culture conditions. We hypothesized that T cells and neuronal cells release cytokines during their interactions, inducing more inflammation via autocrine and paracrine effects. We performed ELISA analysis of supernatants collected from the in vitro culture and ex vivo brain slices within 24 h of co-culture. We compared neuronal cells alone, neuronal cells with T cells, neuronal cells with the cytokine cocktail, and neuronal cells with T cells and the cytokine cocktail, the condition used for the T cell migration experiments. The supernatant collected from the first 10 min of co-culture was used as the baseline, as the culture medium was changed before sampling. We applied this cytokine mixture to the neuronal culture and brain slices to trigger an inflammatory reaction and observed how the cells reacted to this cytokine stimulus. 

The neuronal cells alone showed no detectable cytokine release within 24 h. However, when co-cultured with T cells, the culture started to produce both IFNγ (12 h: 137.9 ± 23; 18 h: 297.4 ± 26; and 24 h: 577.0 ± 9 pg/mL) and TNFα (12 h: 10.8 ± 3; 18 h: 32.1 ± 7; and 24 h: 101.8 ± 46 pg/mL) after 12 h. Furthermore, when supplemented with the cytokine cocktail in the neuronal culture, pronounced IFNγ (10 min: 56.6 ± 6; 6 h: 207.4 ± 27; 12 h: 450.1 ± 11; 18 h: 716.1 ± 22; and 24 h: 873.5 ± 42 pg/mL) and TNFα (6 h: 26.1 ± 8; 12 h: 52.3 ± 11; 18 h: 102.6 ± 19; and 24 h: 154.1 ± 36 pg/mL) were detected within 24 h. The co-culture supplemented with cytokines produced the highest amount of both IFNγ (10 min: 78.1 ± 7; 6 h: 444.7 ± 34; 12 h: 625.3 ± 18; 18 h: 880.3 ± 29; and 24 h: 995.6 ± 31 pg/mL) and TNFα (6 h: 42.1 ± 5; 12 h: 97.8 ± 23; 18 h: 132.4 ± 30; and 24 h: 224.8 ± 20 pg/mL) among all groups (Figure 4A). The release kinetics were fitted for all cases with both a linear regression function and a single exponential growth function. The R² values were calculated as a measure of quality (Appendix A). Based on R² value analyses, a linear fit can better describe IFNγ release, while a single exponential fit was better suited to describe TNFα release (Figure 4; Appendix A). Of note, TNFα release in neuronal culture did not progress exponentially, possibly due to the marginal presence of microglia in the culture.

We observed a linear increase of up to 14 fold IFNγ release within 24 h of T cell–neuronal co-culture with the cytokine cocktail supplement, whereas in the case of TNFα, the cytokine was generated 6 h later and showed exponential growth to 12 fold (Appendix A). The T cells contributed to additional cytokine production when co-culturing with neurons. The greatest difference occurred at 6 h (approximately 237 pg/mL), and the difference decreased over the next 18 h (approximately 122 pg/mL). TNFα production was greater in the culture with added T cells (Appendix A). These results are consistent with T cell–neuron interaction, which enhances signaling. The addition of T cell-conditioned medium triggered an inflammatory reaction in the co-culture. With the initial cytokine trigger, the co-culture produced 5.1 times more IFNγ at 12 h and gradually decreased to 3.7 times at 18 h and 1.7 times at 24 h. TNFα production followed a similar trajectory. The TNFα concentration was 12.3 fold greater in the first 12 h and dropped to 3–4 fold within 24 h under these conditions (Appendix A).

For the ex vivo brain slices, the proinflammatory cytokine profile was similar to that of the in vitro condition. The cytokines were not detectable in the supernatant of the brain slices alone. After cytokine cocktail application, the cultures contained a significant amount of IFNγ (10 min: 139.9 ± 16; 6 h: 639.8 ± 44; 12 h: 821.9 ± 56; 18 h: 1029.4 ± 42; and 24 h: 1260.1 ± 108 pg/mL) and TNFα (10 min: 25.5 ± 8; 6 h: 27.3 ± 10; 12 h: 64.8 ± 11; 18 h: 129.0 ± 29; and 24 h: 368.5 ± 90 pg/mL). Brain slices co-cultured with T cells in ACSF also produced IFNγ (6 h: 71.1 ± 30; 12 h: 131.1 ± 84; 18 h: 368.8 ± 48; and 24 h: 708.4 ± 54 pg/mL) and TNFα (6 h: 9.6 ± 7; 12 h: 36.8 ± 20; 18 h: 69.3 ± 34; and 24 h: 132.9 ± 59 pg/mL) from the first 6 h. The co-culture had the highest cytokine levels when supplemented with the cytokine cocktail. IFNγ (10 min: 164.5 ± 23; 6 h: 788.7 ± 94; 12 h: 981.5 ± 87; 18 h: 1225.0 ± 102; and 24 h: 1389.6 ± 144 pg/mL) and TNFα (10 min: 43.0 ± 11; 6 h: 71.8 ± 25; 12 h: 102.3 ± 18; 18 h: 190.9 ± 40; and 24 h: 579.0 ± 75 pg/mL) increased within 24 h (Figure 4B). Interestingly, despite the change in brain cell composition in the brain slices compared to the in vitro neuronal culture, IFNγ showed a linear increase up to 8.6 fold within 24 h, while TNFα increased exponentially, with a 17 fold change (Appendix A). Furthermore, the addition of T cells increased IFNγ to near 150 pg/mL at all measured time points, while an increase in TNFα (129.6 pg/mL) was detected at 24 h of co-culture. Finally, consistent with the in vitro data, the addition of cytokines to the T cell–brain slice co-culture led to a 4.9 fold increase in IFNγ at 12 h in contrast to the co-culture without the cytokine cocktail; this value slowly decreased in the next 18 h (Appendix A).

These results confirm that IFNγ and TNFα were released in our in vitro and ex vivo co-culture models. The cytokine release profiles were similar in the neuronal culture and brain slice culture. The addition of T cells increased cytokine levels, indicating an interaction between the T cells and brain cells. Furthermore, these results indicate that there were inflammatory conditions in both preparations.

## 3. Discussion

Multiple sclerosis is an autoimmune disease of the CNS. Autoreactive T cell infiltration into the CNS contributes to inflammation and subsequent pathology. This process can be mimicked by EAE, an animal model of MS. Different EAE models, including mouse strains, transferred encephalitogenic T cell clones, and transgenic recipient mouse lines, phenocopy different aspects of MS. In MS, lesions appear in both white and gray matter. To understand the cortical damage caused by autoreactive T cells, we established in vitro and ex vivo T cell–neuron inflammatory models and analyzed how T cells responded to the inflamed neuronal network. We visualized CD8+ T cells migrating on the neuronal network, as well as within neocortices, by live confocal imaging. Contact and interaction between T cells and cortical neurons can be observed in the in vitro and ex vivo models (Figure 3A,F). In addition, reduced migration velocity was detected under inflamed conditions in both models (Figure 3), consistent with in vivo observations [31,39].

Although T cell–neuron co-culture models have been used previously [40,41], here we applied systematic in vitro and ex vivo preparations to study the interactions of T cells with neurons during inflammation. These models are easily adaptable and manipulatable. For instance, as many infiltrating immune cell populations have been identified at the lesion sites of MS patients, one could investigate neuronal interactions with target immune cell types (i.e., B cells or NK cells) by applying them to neuron cultures or brain slices. To investigate specific inflammatory pathways affecting neurons or glia cells, one could control the cytokine concentration or cytokine cocktail combination to uncover the complexity of the inflammatory response by mimicking chronic or acute neuroinflammation. The models we provide here offer significant accessibility for analyzing and monitoring various cellular activity parameters of T cells and neuronal cells, such as changes in calcium signaling, synaptic transmission features, and T cell or glia cell activation under neuroinflammatory conditions. Immune cell activation plays a part in modulating the acute and chronic progression of neurodegenerative diseases [42]. Despite the strength of the in vitro and ex vivo model combination, a lack of physiological integrity remains when studying neurological–autoimmune pathology. The unmodified crosstalk between the immune system and CNS is preserved solely in the living organism. Therefore, in vivo studies are still indispensable. It is important to verify the findings from in vitro and ex vivo studies using in vivo models. 

Isolated brain cells from the neocortex of postnatal day 0 pups consisted mainly of cortical neurons. Due to a lack of oligodendrocytes, myelin was absent in our neuronal culture. Therefore, we generated polyclonal activated effector CD8+ T cells without priming them to neural antigens for the in vitro study. We optimized the co-culture conditions by using Neuronal-A culture medium due to the susceptibility of neurons. Activated CD8+ T cells appeared to tolerate the NBA medium well in the co-culture. For the ex vivo study, we generated neural antigen-reactive T cells by priming naive T cells from splenocytes with the MOG_35-55_ peptide, the peptide clone that can induce EAE in C57BL/6 mice. When co-culturing ex vivo brain slices, we immersed the brain slices and T cells in standard artificial cerebrospinal fluid, which was also well tolerated by the T cells. The quality of the brain slice is decisive for fine neuronal activity measurement. We cut acute brain slices in ice-cold high sucrose buffer using a vibratome to minimize neuronal damage. The use of ice-cold buffer for brain slicing has been applied extensively for performing patch clamping [43,44,45], the most sensitive method for measuring neuronal activity. Eguchi et al. compared ultrastructural and electrophysiological features of synapses in mouse acute brain slices prepared at ice-cold and physiological temperatures (35–37 °C). The results pointed out that preparation under physiological temperature provides substantial advantages for investigating synaptic functions [46]. It might be worthwhile to test whether acute brain slices prepared under physiological temperature better preserve brain cellular activities, which consequently fine tune the regulation of the T cell attack in the co-culture system.

Recombinant proinflammatory cytokines, such as IL-1β, IFNγ, and TNFα, are often used to produce neuroinflammation [40,41,43]. These cytokines generate an inflammatory response with changes in neuronal calcium signaling, electrical activity, and T cell migration. Inflammation consists of a tightly regulated process associated with changes in the cytokines produced by cells. It is unclear which cytokines occur at the site of pathology and what kind of effects they contribute locally and at other locations. To mimic the inflamed brain situation, instead of adding solely proinflammatory cytokines IFNγ and TNFα, we added conditioned medium produced by CD8+ T cells during their polyclonal activation. As autoreactive T cells undergo reactivation via contact with leptomeningeal phagocytes shortly before entering the CNS [9], we bypassed this reactivation process by utilizing preactivated effector T cells to neuronal culture or brain slices. 

We visualized real-time T cell migration on cortical neurons and in neocortical brain slices. Under inflamed conditions, CD8+ T cells drastically reduced their migration velocity and altered their migration pattern. This altered migration appeared in both the in vitro and ex vivo models (Figure 3), which is in good agreement with the behavior of neural antigen-reactive infiltrating T cells in the inflamed brains of EAE animals [31,39]. Reduced T cell migration speed implies antigen recognition and cell activation through synapse formation. MHC I expression is upregulated in brain cells followed by IFNγ exposure [1,2], thereby increasing the likelihood that CD8+ T cells will recognize MHC I- expressing cells. Furthermore, CD8+ T cells formed synapses upon MHC I molecule recognition and attacked their cognate target cells [44]; supplemented cytokines might support the activation of T cells. Although antigen-specific contacts of myelin protein-primed T cells to neurons contribute to reduced migration speed in contrast to antigen-unprimed T cells [45], the supplemented cytokine appears to overcome the effects and change the behavior of effector T cells on neuronal cells and within neocortices (Figure 3). Notably, MHC I molecules have been reported to play a role in synaptic plasticity and regeneration of neurons after nerve damage [44,46]. Since CD8+ T cells play a double-sided role in neuronal damage and neuroprotection, it would be interesting to know whether CD8+ T cells also contribute to neuroregeneration via MHC I engagement signaling.

Neuroinflammation can be intrinsic and lead to blood–brain barrier leakage and infiltration of T cells. In this scenario, autoreactive T cells enter the inflamed CNS before they contribute their proinflammatory cytokines. Glia cells, which maintain and support neuronal function, release a broad spectrum of inflammatory cytokines when challenged. However, brain cells react to the culture environment, and isolated glia cells tend to be activated in the cell culture, as well as in the sliced brain. Increased glial fibrillary acidic protein (GFAP) expression in astrocytes indicating astroglial activation was observed in the neuronal culture and acute brain slices (Figure 1C), similar to that following brain injury or stress [33]. We examined how CD8+ T cells responded in inflamed conditions and how long the supplemented cytokine effects remained in the neuronal culture. T cells showed dramatically reduced migration velocity in the 12 h pre-incubated neuronal culture (Figure 3D,E), but their velocity recovered during the 24 h pre-incubation. We speculate that neuronal cells, in response to cytokines released from T cells, express MHC I or other inflammatory-driven upregulated molecules, such as intercellular adhesion molecule-1 (ICAM-1) [47,48]. ICAM-1 is the counter ligand of lymphocyte function-associated antigen-1 (LFA-1) [49]. It is known to play essential roles in inflammatory processes and in T cell-mediated host defense mechanisms, resulting in decreased T cell migration velocity. After 24 h of inflammation, neurons may respond by reducing this response (e.g., by reducing MHC I molecule expression), resulting in a slight increase in T cell migration speed compared to controls. Overall, the migration data indicate that our in vitro and ex vivo models allowed for the study of T cell–neuronal interaction and the molecular basis of neuroinflammation in 2D and 3D neuronal networks.

Finally, our ELISA data verified production of the two key proinflammatory cytokines, IFNγ and TNFα, in both co-culture models within 24 h. It is surprising that when supplemented with cytokines in the absence of T cells, neuronal cells and brain slices are able to release high amounts of IFNγ and TNFα. Additional cytokines were produced when T cells were introduced. These data reveal the fact that activated T cells induced neuroinflammation through cytokine release and interaction with resident brain cells, which leads to synergistic effects on cytokine secretion. Moreover, the increases in cytokines were similar in the neuron culture and brain slices when co-cultured with T cells. IFNγ showed a linear increase, while TNFα showed an exponential increase in both co-cultured models (Figure 4 and Appendix A). Astrocytes (62%) were the main population, followed by neurons (35%), in the in vitro culture, while neurons (72%) were the dominant cell population in the neocortices, followed by astrocytes (13%) (Figure 1). We did not expect such consistency in these responses since microglia, the main TNFα producers, were notably underrepresented in the in vitro neuronal culture. Despite the differences in cell composition and neuronal network structure, it appears that neuron–glia networks can generate additional cytokine production, possibly to retain homeostasis against cell stress. The ELISA data revealed that our in vitro and ex vivo brain slice models responded in a similar fashion to neuroinflammation.

In this study, we established in vitro and ex vivo co-culture models mimicking neuroinflammatory conditions to study T cell and neuronal interaction. We verified the inflamed condition in both models by detecting the release of two key proinflammatory cytokines (IFNγ and TNFα). These inflamed neuronal networks resulted in a reduced T cell migration phenotype, which is consistent with the phenotype in the inflamed brains of EAE animals from other studies. Our in vitro and ex vivo models provide a promising platform for studying the molecular details of neuronal and immune cell interaction under inflammatory conditions, which is amenable to manipulation of numerous experimental factors.

## 4. Materials and Methods

### 4.1. Mice

Wild-type (WT) postnatal day 0 or day 1 pups were used for neuronal culture. The 2–3-week-old transgenic mice TgN (Thy1.2-HcRed)_THRE_ containing far-red fluorescent protein HcRed1-expressing neurons, driven by the Thy1.2 promoter [50], were used to prepare the acute brain slices. The 8–20-week-old CD8cre1-τGFP mice were used to acquire T cells. CD8cre-τGFP mice were generated by crossing a previously published reporter τGFP line with a CD8a-Cre line (Jackson Laboratory, #008766). The WT mice and transgenic mice used in this study were all in C57BL/6N background. Both female and male mice were used in the experiments. The animals were kept under housing conditions of 22 °C room temperature (RT) with 50–60% humidity and 12 h dark/light cycles. All experimental procedures were approved and performed according to the regulations of the state of Saarland (Landesamt für Verbraucherschutz, AZ.: 2.4.1.1). All animal experiments were performed according to German law and European directives and with permission from the state of Saarland (Landesamt für Gesundheit und Verbraucherschutz; animal license number 41-2016, Perfusion (FKI/2020)). 

### 4.2. Plasmid and Lentivirus Production

Neuron-specific infection pCDH-hSyn vectors were amplified from the backbone vector pZac2.1 hSynapsin1 NAPA-N2 SV40 (addgene #97212) using the forward primer 5′-ATGTATAGATATCAGTGCAAGTGGGTTTTAGGACCAG-3′ and reverse primer 5′- ATGTATACGCGGATCCCTGCGCTCTCAGGCACGACAC-3′ with restriction sites EcoRV and BamHI. The amplified hSyn product was then subcloned into a pCDH-EF1 (Addgene #72266) lentiviral transfer vector by replacing the elongation factor-1 alpha promoter (EF1α). The fluorescent protein mScarlet I was subcloned into a pCDH-hSyn vector with XbaI and XmaI restriction sites. The cDNA of the 4x mitochondrial targeting signal (4xmts) with the restriction sites NotI and PstI was purchased from the company BioCAT in pBluescript II SK(+) vector. The insert 4xmts cDNA was cloned into a pCDH-hSyn-mScarlet I vector. All constructs were verified with DNA sequence analysis (Microsynth Seqlab, Göttingen, Germany).

Lentivirus was produced using the packaging cell line Lenti-X293 (cat. 632180; Takara, Shiga, Japan) as described by the manufacturer. Briefly, 11 × 10^6^ Lenti-X293 cells in a 10 cm dish were starved in DMEM medium without FCS supplement for 5 h before calcium phosphate transfection. Buffer (2x HBS) containing 280 mM NaCl, 50 mM HEPES, and 1.5 mM NaH_2_PO_4_ was added dropwise into a calcium-based plasmid mix solution (1.8 mL sigma water + 200 μL 2.5 M CaCl_2_ solution) containing 180 μg lentiviral-expressing plasmid (pCDH-hSyn-4xmts-mScarlet), 80 μg pMD2.G (Plasmid #12259; Addgene), 80 μg pMDLg/pRRE (Plasmid #12251; Addgene), and 80 μg pRSV-Rev (Plasmid #12253; Addgene). The plasmid mixed solution was incubated for 25 min at RT before application to the starved Lenti-X293 cells. The lentiviral particles were harvested 2.5 days after transfection. Finally, the viral particles were collected, followed by ultracentrifugation for 2.5 h at 22,500 rpm at 4 °C. 

### 4.3. Acute Brain Slice Preparation

Thy1.2-HcRed mice were sacrificed by CO_2_ inhalation and decapitated. Their brains were quickly removed and placed into ice-cold high sucrose buffer containing (in mM): 230 Sucrose Ultra, 2 KCl, 1 KH_2_PO_4_, 10 D-(+)-glucose monohydrate, and 26 NaHCO_3_. Coronal brain slices (350 mm thick) were cut using a vibrating microtome (Leica VT1200 S) and transferred using a glass Pasteur pipette to a recovery chamber containing artificial cerebrospinal fluid (ACSF). The ACSF contained (in mM): 120 NaCl, 2 KCl, 1.3 MgCl_2_, 1 KH_2_PO_4_, 2 CaCl_2_, 26 NaHCO_3_, and 10 D-(+)-glucose monohydrate and was saturated with carbogen (95% O_2_ and 5% CO_2_ mixture; pH 7.4). The brain slices were recovered in the carbogenated ACSF at 35 °C for at least 2 h before the experiments.

### 4.4. Cell Culture

#### 4.4.1. T Cell Culture for In Vitro Study 

Splenocytes were isolated from the spleens of CD8cre-τGFP mice followed by smashing through a 70 µm cell strainer in cold RPMI medium supplemented with 10% FCS and 1% penicillin/streptomycin (P/S). The splenocytes were collected after removing erythrocytes from the cell suspension using a water-based lysis buffer containing 155 mM NH_4_Cl, 10 mM KHCO_3_, and 0.13 mM EDTA. An amount of 1 mL lysis buffer was applied to the splenocyte pellet from one spleen for 30 s at RT. Afterward, 9 mL RPMI medium was supplied to the 15 mL falcon to stop the lysis reaction. After centrifuging the cells at 300× *g* for 6 min, the cell suspension was further washed once with D-PBS containing 2 mM EDTA and 0.1% BSA before CD8 positive isolation. Naive CD8+ T cells were positively isolated from the splenocytes using a Dynabeads FlowComp Mouse CD8+ kit (Invitrogen) as described by the manufacturer. The isolated naive CD8+ T cells were stimulated with anti-CD3/anti-CD28 activator dynabeads (1:0.8 ratio; Invitrogen) and cultured for 5 days. The cells were cultured at a density of 1 × 10^6^ cells/mL in a 24-well culture plate (2 million cells per well) with AIMV medium (Invitrogen) containing 10% FCS, 1% P/S (Invitrogen), and 50 μM 2-mercaptoethanol (BME). Afterward, 50 U/mL recombinant mouse IL-2 (Gibco) was applied in the first 2 days of T cell culture. Then, 100 U/mL mouse IL-2 was added after 48 h of culture to support T cell proliferation. These activated effector CTLs were used for the neuronal cell co-culture study. 

#### 4.4.2. MOG-Specific T Cell Culture

To generate MOG-specific T cells, naive splenocytes isolated from WT mice or CD8cre-τGFP transgenic mice were incubated with 75 µg/mL MOG_35-55_ peptide (Genaxxon) in RPMI culture medium containing 50 µM BME, 10% FCS, and 1% P/S that was supplemented with 0.5 µg/mL anti-CD3e (BD Pharmingen, clone 145-2C11, cat. 557306), 0.2 µg/mL anti-CD28 antibody (BD Pharmingen, cat. 553294), and 50 U/mL mouse IL-2 for 2 days. The splenocytes from one spleen were cultured in 5 mL of culture medium in one well of a 6-well plate. Blast cells were generated after 2 days (day 2) in culture. Then, 100 U/mL of IL-2 was added to the culture medium at day 3 to induce the proliferation of T cells. The T cells were separated from mixed splenocytes at day 3 or day 4 using Lymphocyte Separation Medium 1077 (PromoCell) with Ficoll density gradient centrifugation at 400× *g* for 30 min without break at RT. The purified T cells (1 × 10^6^ cells/mL) were further cultured in RPMI culture medium supplemented with 100 U/mL IL-2 for 4 days (day 6). To expand the T cell population, we repeated the T cell priming protocol as with naive splenocytes but using thymocytes from another WT mouse with C57BL/6N background as antigen-presenting cells and restimulated the T cells on day 5 or day 6 with 75 µg/mL MOG peptide (second restimulation, day 0). It is notable that in the second cycle of stimulation, we did not provide suspended anti-CD3/anti-CD28 antibodies because these T cells were activated previously. The T cells formed blast cells after 2 days of priming. We utilized these cells after T cell purification with Ficoll density gradient centrifugation on day 4 for brain slice co-culture.

Thymocytes were isolated from the thymus by smashing through a 70 µm cell strainer in cold RPMI medium with 10% FCS. The thymocytes were collected to co-culture with day 5 or day 6 T cells (thymocyte:T cell ratio = 20:1) for restimulation. Five million T cells and 100 million thymocytes were co-cultured in one well of a 6-well plate in RPMI culture medium for 2 days, as in the stimulation protocol described above. Day 4 and day 5 MOG-specific T cells after thymocyte restimulation were used for cell migration studies with brain slices after lymphocyte purification.

#### 4.4.3. Primary Cortical Neuron Culture 

Cortical neurons were isolated from postnatal day 0–1 WT pups. Briefly, the cortices were isolated from the whole brain in ice-cold Earle’s Balanced Salt Solution (EBSS, Gibco). After peeling off the meninges, the cortices were digested with 35 units/mL papain (Worthington, NJ) for 45 min at 37 °C, followed by gentle mechanical trituration. Single cell suspensions were seeded on 25 mm glass coverslips placed in 6-well culture plates (3 × 10^5^ cells per coverslip). The glass coverslips were precoated with a mixture of coating solution containing 17 mM acetic acid, poly-D-lysine (Sigma, St. Louis, MI, USA, P6407), and collagen I (Gibco, A1048301). The neurons were cultured in Neuronal-A (NBA) medium containing 10% FCS, 1% P/S, 1% GlutaMAX, and 2% B-27 supplement (Gibco) for 8–14 days before the experiments. The culture medium was replaced with fresh medium on the second day (day 2) to remove the remaining cell debris from the cell preparation. Afterward, neurons stayed in the conditional medium throughout the experiments, unless additional supplements are mentioned. To label mitochondria in the neurons, the neuronal cells were infected with lentivirus for 20 h in NBA medium without antibiotics on the next day of cell seeding. After 20 h, the virus was removed by replacing with fresh NBA culture medium. The neurons started to express mitochondrial markers 5 days after infection.

#### 4.4.4. T Cell–Neuron Co-Culture for In Vitro Study 

Day 3–4 bead-activated CD8+ T cells were used to co-culture with up to 8-day-old neuronal cells. The T cells (0.5 × 10^6^) were added to each neuronal culture well containing 2 mL NBA culture medium for co-culture. To induce an inflamed condition, 50 µL T cell conditioned medium was added to each neuronal coverslip containing 2 mL of NBA medium. 

#### 4.4.5. T Cell–Brain Slice Co-Culture for Ex Vivo Study 

Coronal brain slices were further cut sagittally into half pieces before co-culturing with T cells. MOG-specific T cells (2 × 10^6^) were applied to one half brain slice in ACSF. Four half brain slices were placed in one well of a 6-well plate with a total of 8 × 10^6^ T cells per well. To induce an inflamed condition, 150 μL of T cell conditioned medium was added to each well containing 2 mL of ACSF. 

Co-cultured conditions were used for the T cell migration assay as well as ELISA analysis. Further technical details will be described in the sections below. All the cell cultures were controlled at 37 °C with 5% CO_2_, and brain slices were carbogenated with a 95% O_2_ and 5% CO_2_ mixture at 35 °C. 

### 4.5. Confocal Imaging

For the live T cell migration in vitro assay, day 3 or day 4 activated CD8-τGFP T cells were applied to an imaging chamber containing neuronal cells in 2 mL NBA conditional culture medium (before cytokine application). T cell migration was recorded using a confocal microscope (LSM 780, Zeiss, Germany). After 45 min of recording, 50 μL of T cell conditioned medium (cytokine cocktail) was applied to mimic inflamed conditions. Afterward, another 45 min was recorded (after cytokine application). For T cell migration on brain slices (ex vivo assay), image recording was first started 2 h after the MOG-specific T cells were applied to the brain slices in order to visualize penetrating T cells in the brain slices. An amount of 150 µL T cell supernatant was applied to four pieces of half-brain slices in 2 mL ACSF to create the inflamed condition. Live imaging was performed at 37 °C for the in vitro assay or at 35 °C for the ex vivo assay with a stable carbogen supply (95% O_2_ and 5% CO_2_ gas mixture). Image acquisition information is stated in each figure, including z-stack and time resolution. The images were acquired with a 40x Plan-Apochromat objective (NA 1.4). For live and fixed samples, τGFP and anti-NeuN-Alexa488 fluorescence was excited at a 488 nm excitation wavelength, anti-GFAP-Alexa568 was excited at 568 nm, and anti-Iba1-Alexa647 was excited at 647 nm. For the brain slices, the image stacks were acquired as 12 µm total thickness with a 2 µm interval distance. For the cultured cells, 6 µm total thickness with a 1 µm interval distance was acquired. The maximum intensity projection images are shown in the live-cell imaging data. 

### 4.6. ELISA

For ELISA analysis, supernatants were collected after 10 min, 6 h, 12 h, 18 h, and 24 h from T cell–neuron co-culture or T cell–brain slice co-culture. The concentrations of IFNγ and TNFα were quantitatively measured using an IFNγ ELISA kit (abcam, ab46081) and a TNFα ELISA kit (abcam, ab208348) as described by the manufacturer. Briefly, 100 μL of 1:10 diluted culture supernatants or 50 μL of 1:2 diluted culture supernatants were added to the IFNγ antibody or TNFα antibody precoated 96-well plates for 1 h, respectively, to form an antibody–antigen complex. After the washing steps, 100 μL TMB substrate was applied to the wells for 30 min at RT in the dark. Sequentially, 50 μL of stop solution was added to stop the reaction. Finally, the samples were read (TECAN; infinite M200 pro plate reader) at 450 nm wavelength. Four independent co-culture preparations were performed in both cytokine measurements. 

### 4.7. Immunofluorescence

For brain slice staining, p12 WT mice were deeply anesthetized using an intraperitoneal injection of a mixture of ketamine (280 mg/kg bodyweight) and xylazine (20 mg/kg bodyweight). The mice were intracardially perfused with D-PBS for 5 min, followed by a fixative containing 4% paraformaldehyde for another 5 min. After perfusion, the brains were isolated and post-fixed in the same fixative for another 24 h. After washing once with PBS, the brains were sliced into 50 µm sections using a vibratome. The fixed brain slices were washed once with PBS and permeabilized with 0.1% triton (permeabilization buffer) for 10 min, followed by blocking with an extra 10% FCS in permeabilization buffer for 30 min at RT. Afterward, the samples were incubated overnight at 4 °C with primary antibodies anti-NeuN (1:200; Merck, clone A60), anti-GFAP (1:200; Abcam, ab4674), and anti-Iba1 (1:200; Wako, Code No. 019-19741). After washing three times with permeabilization buffer, corresponding secondary Alexa fluorophore-coupled antibodies were further incubated with the brain slices for 1 h at RT. Finally, the samples were washed three times with D-PBS, followed by DAPI (1:1000) staining and mounted with Fluoromount-G Mounting Medium (Invitrogen). For neuronal cell staining, the cells were fixed on day 12 with 4% PFA for 10 min at RT. After washing three times with D-PBS, the cells were stained following the staining protocol as for the brain slices except using 2% BSA for blocking. 

For the MOG-specific lymphocyte staining shown in Figure 2C, second stimulated day 4 cells (0.2 × 10^6^) isolated from WT mice were collected from the culture in 1.5 mL Eppendorf tubes. The cells were resuspended in D-PBS and stained with anti-CD4-APC (1:200; eBioscience, GK1.5) and anti-CD8a-PE-CF594 (1:200; BD Horizon, Clone 53-6.7) antibodies for 45 min on ice. Afterward, the cells were spun down with 300 *g* for 5 min and washed once with 1 mL of D-PBS to remove unbound antibodies. Finally, the cells were resuspended again in 300 μL of D-PBS for confocal imaging.

### 4.8. Statistics and Image Analyses

The Mann–Whitney U test was used to compare values for T cell migration analysis. Values were considered statistically significant when probability (P) values were below 0.05 (*), 0.01 (**), or 0.001 (***). Data were analyzed with ImageJ v1.46 [51], Excel (part of Office 2013, Microsoft), SigmaPlot 14, and Imaris 9.3 (Bitplane AG) and graphed using Affinity Designer Software (Serif Ltd.). Fitting analysis for IFNγ and TNFα release as a function of time was performed with SigmaPlot 14 using linear regression and a single exponential growth function. 

For T cell migration analysis, confocal live images were exported to Imaris software to analyze CD8+ T cell migration co-cultured with neuronal cells. T cell movements were tracked using the Imaris spot object-based algorithm on source channel 488 to create tracks for each cell. Due to the uneven cell migration duration and tracks in the field of view, we included only the tracks dwelling over 15 min of the recordings in order to acquire a more accurate mean migration speed. All cell tracks were manually corrected using the manual editor when misdetection, nondetection, or false cell duplicates were observed. Some smaller debris or dead cells were detected by the software, despite the use of a threshold of 5–6 uM. Additionally, overlapping cells resulting in false single-track detection were manually corrected. Overall, with background subtraction and the set threshold mentioned above, we set the quality threshold for CD8+ T cell selection to 15–20% in our analysis. The parameters collected from migration analysis were defined as follows: Track duration (min): duration between the first and last time point within the tracks; Track length (μm): total length of the measured track; Track speed/velocity mean (μm/min): track length divided by the time between the first and last object within the track; Displacement (μm): straight distance between the first and last cell positions; and Track straightness: displacement divided by track length.

## Figures and Tables

**Figure 1 ijms-24-03166-f001:**
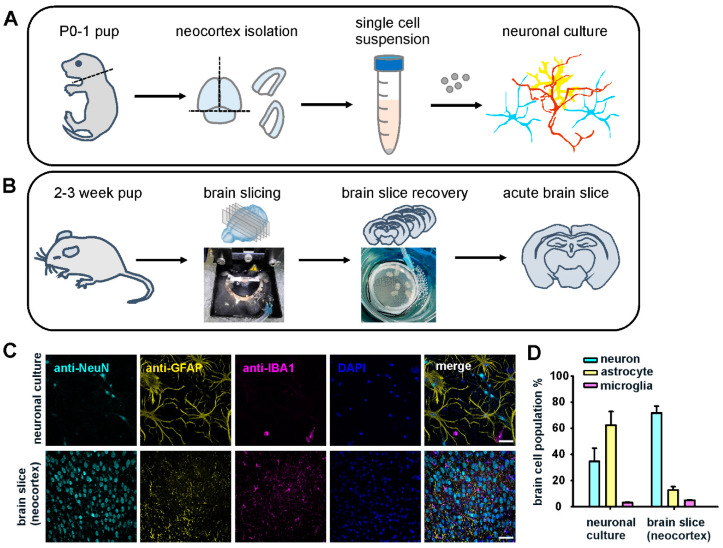
Schematic illustration of neuronal culture and acute brain slice preparation and brain cell composition in cell culture and neocortex. (**A**) Neuronal cells were isolated from the neocortices of postnatal day 0 or day 1 WT pups. A single-cell suspension was obtained after tissue digestion with papain. Finally, the cells were seeded on collagen-coated glass coverslips and cultured for 8–12 days before use. (**B**) Schematic illustration of acute brain slice preparation. Whole brains were isolated from 2- to 3-week-old pups. Brain slices were cut using a vibratome in ice-cold high sucrose buffer. The brain slices were transferred to artificial cerebrospinal fluid (ACSF) for tissue damage recovery before the experiments. (**C**) Confocal images of neuronal cells in neuronal culture and neocortex from (**A**,**B**). Day 8 neuronal cells and acute brain slices after recovery were fixed and stained with anti-NeuN, anti-GFAP, and anti-IBA1 antibodies to label neurons, astrocytes, and microglia, respectively. Cell nuclei were labeled with DAPI. Scale bar: 50 µm. (**D**) Quantitative analysis of brain cells in cell culture and neocortex from (**C**). Cell counting analysis in both the cell culture and brain slice groups was performed from two independent preparations (N = 2; *n* = 1249 for cell culture and *n* = 1036 for brain slice).

**Figure 2 ijms-24-03166-f002:**
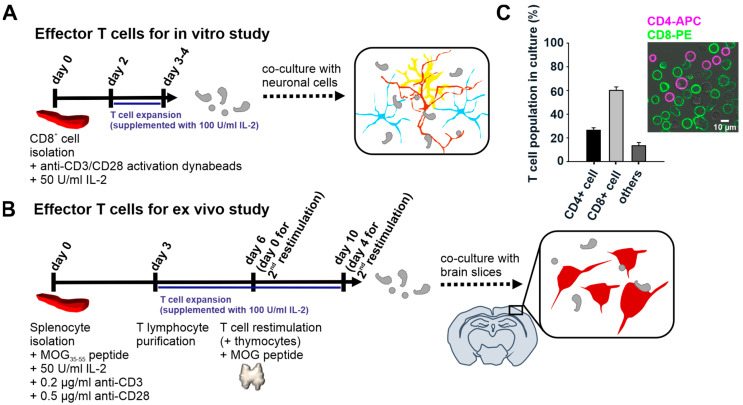
Effector T cell generation for co-culturing with neuronal cells or brain slices. (**A**) Workflow of effector T cell generation for the in vitro study. Naive CD8+ T cells were positively isolated from splenocytes and were activated by anti-CD3/anti-CD28 activation dynabeads. The T cell culture was supplemented with mouse IL-2 protein for further proliferation. The cells were used for co-culture with neuronal cells on day 3 or day 4 culture. (**B**) Workflow of MOG antigen-specific effector T cell generation for the ex vivo study. Naive T cells were cultured in a splenocyte pool that was supplemented with MOG peptide and IL-2 protein to prime T cells with the CNS-specific antigen. After 3 days in culture, the T cells were purified using Ficoll density gradient centrifugation. After 6 days in culture, the T cells were further restimulated with thymocytes supplemented with MOG peptide (second restimulation). Finally, blast cells were developed in culture 2 days after restimulation (day 8). Day 4 T cells (second restimulation) were used for co-culturing with the acute brain slices. (**C**) Quantitative analysis of the MOG-specific T cell population generated from WT mice in culture from (**B**). A confocal image of MOG-specific lymphocytes stained with anti-CD4-APC and anti-CD8-PE antibodies. Cell counting analysis was conducted from three independent preparations (data presented as mean ± SEM; N = 3; *n* = 6388).

**Figure 3 ijms-24-03166-f003:**
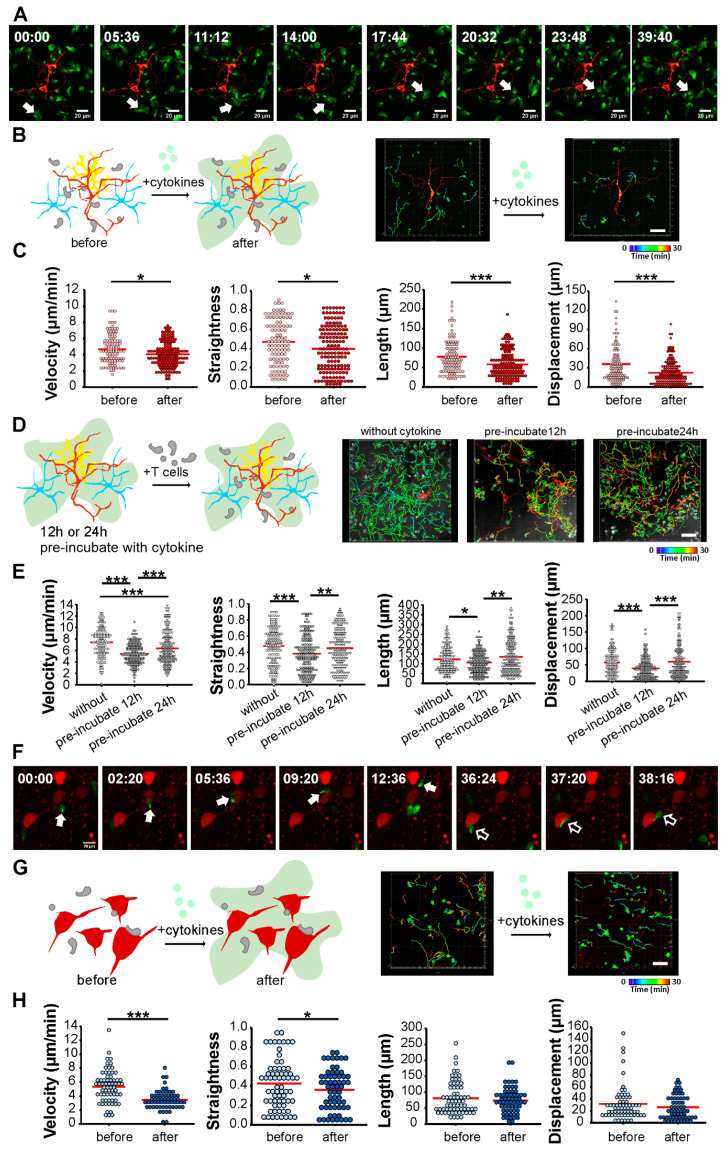
T cell mobility is altered under inflammatory conditions in both in vitro culture and ex vivo brain slices. (**A**) Confocal images of CD8cre-τGFP T cells (green) migrating on a neuronal layer. CD8cre-τGFP T cells (green) were co-cultured with cortical neuronal cells expressing mitochondrial marker cox8a-mscarlet (red). White arrows point to migrating T cells. (**B**) T cell migration was compared before and after cytokine application on neuronal culture. Left: scheme of the experimental design correlating to (**C**). Right: confocal snapshots from live images of T cells migrating on neuronal slice cultures. T cell migration tracks are shown in rainbow color gradient correlating to 30 min recordings. (**C**) Quantitative analysis of velocity, straightness, track length, and displacement of CD8-τGFP cells migrating on neuronal cells from (**B**). Analyses were performed with three independent preparations (N = 3; *n* = 143 for the before group and *n* = 148 for the after group). (**D**) Neuronal cells were pre-incubated with the cytokine cocktail for 12 or 24 h to mimic the inflamed condition before T cells were applied for co-culture. Scheme of experimental design correlating to (**E**). Right: confocal snapshots of T cell migration on neuronal culture before cytokine application and cytokine pre-incubation. (**E**) Quantitative analysis of velocity, straightness, track length, and displacement of CD8-τGFP cell migration on neuronal cells from (**D**). Analyses were performed with two independent preparations (N = 2; *n* = 163 for the control group, *n* = 193 for the 12 h cytokine pretreatment group, and *n* = 193 for the 24 h cytokine pretreatment group). (**F**) Confocal snapshot images of CD8cre-τGFP T cells (green) migrating in a brain slice in the cortical region. CD8cre-τGFP T cells (green) were co-cultured with a brain slice isolated from a Thy1.2-HcRed mouse in which the neurons expressed HcRed1 fluorescence protein (red). White closed and opened arrows point to two individual migrating T cells. (**G**) T cell migration in brain neocortices was compared before and after cytokine application. Left: demonstrative scheme of experimental design correlating to (**H**). Right: confocal snapshots from live images of T cells migrating on neocortex before and after cytokine application. T cell migration tracks are shown in a rainbow color gradient correlating to 30 min recordings. (**H**) Quantitative analysis of velocity, straightness, track length, and displacement of CD8-τGFP cell migration on neocortices from (**G**). (**B**,**D**,**G**) Red line represents the mean. The Mann–Whitney U test was used to compare values; * *p* < 0.05; ** *p* < 0.01; *** *p* < 0.001. Scale bars: 50 µm.

**Figure 4 ijms-24-03166-f004:**
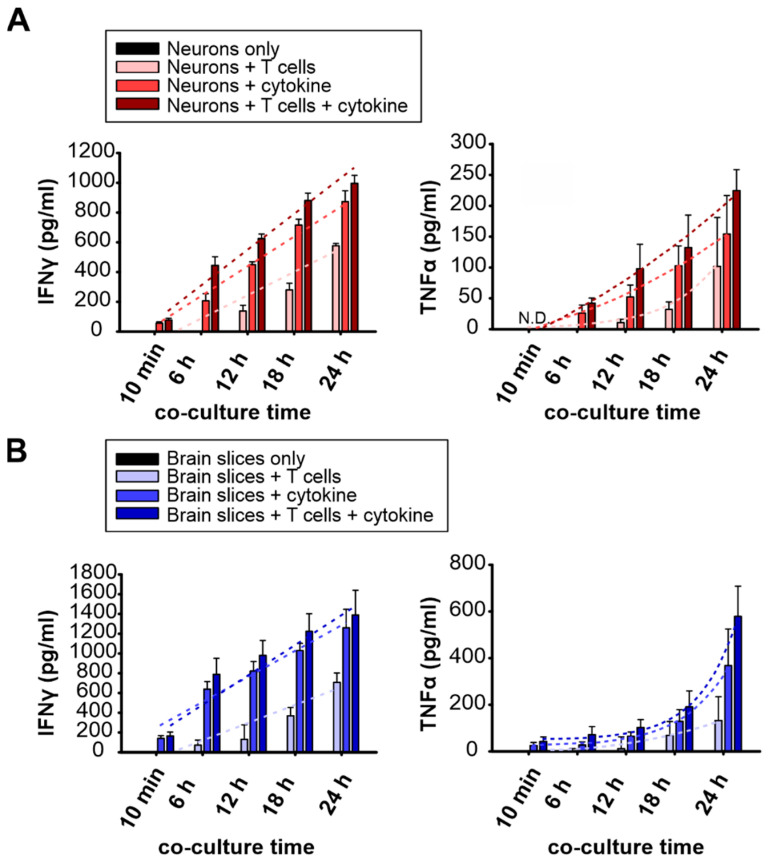
Proinflammatory cytokine release in neuronal culture and brain slices during T cell co-culture. Culture media were collected from (**A**) in vitro T cell–neuron co-culture or (**B**) ex vivo T cell–brain slice co-culture after 10 min, 6 h, 12 h, 18 h, and 24 h. IFNγ data were linearly fit. TNFα data were fit with an exponential growth function (stipple lines). Released IFNγ or TNFα from the different culture conditions were measured by ELISA. ELISA measurements were performed from four independent preparations, with one technical replicate from each preparation. N.D. stands for nondetectable.

## Data Availability

The data presented in this study are available in Appendix A.

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
