# Peer review of "Study of Effector CD8+ T Cell Interactions with Cortical Neurons in Response to Inflammation in Mouse Brain Slices and Neuronal Cultures"

_ijms, 2023, doi:10.3390/ijms24043166_

Round 1
Reviewer 1 Report
The manuscript by Lin et al entitled “CD8+ T cells alter their migratory behavior in response to pro-2 inflammatory cytokines in cortical cultures” is well written and organized. The authors performed in vitro (cortical neuronal culture) and ex vivo (acute brain slice) experiments to determine CD8+T cells interaction with neurons. The goal of this manuscript is to provide detailed methods to perform these experiments using live cell confocal imaging.
Concerns
1. This is a methods manuscript. I believe the title should reflect that. Please, change accordingly.
2. There is a lack of references both in the results and methods sections. I am sure the techniques they used are adaptations to already published ones.
3. There is a strong redundancy between results and methods. Please, move all technical aspects to the methods section.
4. There is a recent paper by Eguchi et al (fncel.2020.00063) where the authors compare cold versus physiological temperatures in brain slices. This should be brought up in the discussion.
5. Figures are not cited in the results sometimes. Please correct.
Author Response
We thank reviewer for the time and efforts in reviewing our manuscript. We appreciate these valuable comments and have addressed these points as follows (labeled in yellow in the manuscript).
Concerns
- This is a methods manuscript. I believe the title should reflect that. Please, change accordingly.
Thanks for this essential comment. We changed the title now to ” Study the interaction of effector CD8+ T cells and cortical neurons in response to inflammation in mouse brain slices and neuronal cultures”.
- There is a lack of references both in the results and methods sections. I am sure the techniques they used are adaptations to already published ones.
We added a small paragraph mentioning the protocol adaptations in the result section (line 149-153, yellow) with multiple references included, concerning protocols of T cells, cortical culture preparation and the co-culture of both.
- There is a strong redundancy between results and methods. Please, move all technical aspects to the methods section.
We have removed most of the technical details of culture preparation in the results and clarified some points in the methods. A simple modified description is shown in the results (line 89-91; line 138-143).
- There is a recent paper by Eguchi et al (fncel.2020.00063) where the authors compare cold versus physiological temperatures in brain slices. This should be brought up in the discussion.
Thanks for this very interesting point. We added a paragraph to discuss the protocol of Eguchi et al in the discussion (line 396-407).
- Figures are not cited in the results sometimes. Please correct.
Thanks for pointing out the editing errors. We have corrected it accordingly.

Reviewer 2 Report
The manuscript by Lin C-H, et al. investigated the effect of proinflammatory cytokines on mouse cytotoxic CD8+ T cell migration in two neuronal co-culture models. The treatment with T cell conditional media decreased CD8+ T cell migration velocity and pattern and further increased the release of INFg and TNFa in both in vitro and ex vivo co-culture models. This study provides a useful tool to study neuron-immune interaction in vitro. However, there are some points that need to be addressed.
Major points:
1. The whole study was conducted only with mouse cells/tissues so it is necessary to include “mouse” in the title.
2. Has the viability of neurons in the brain slice culture been accessed?
3. The majority of cells in the neuronal culture are proliferating astrocytes. Have the authors considered to increase the neuron to glia ratio by applying low concentration of cytosine arabinoside?
4. In the figure 2C, was the T cell subpopulation determined with flow cytometry or immunofluorescent staining? Since MOG-specific T cells are isolated from CD8Cre-GFP mice, why are CD8+ T cell subpopulations not first sorted and further co-culture with brain slices?
5. How will be effect on CD8+ T cell migration when adding IFNg or TNFa alone as compared to the T cell condition media?
6. Figure 4, it might be helpful to fit the data and show the linear or exponential increase.
7. Line 628, the detailed protocol for imaging analysis (migration parameters) is missing.
8. Discussion, the authors may include some discussions on the potential use of these models to study the change of neural activity/transmission/network under neuroinflammatory conditions.
Minor points:
1. What are the error bars in the figure 2? SEM or SD?
2. The colors showing neurons in the figure 2A (blue) and B (red) are different. It is good to keep consistency.
3. On the page 6, there are several places that referred to the wrong figure, e.g., line 207, 227 and 236.
4. Line 257, “affects” should read as “effects”.
5. Line 379, what does “NBA medium” stand for? Neural basal?
6. Line 446, “Microglia” should read as “microglia”.
Author Response
We thank reviewer for the time and efforts on reviewing our manuscript. We appreciate these valuable comments and have addressed these points as follows (labeled in yellow in the manuscript).
Major points:
- The whole study was conducted only with mouse cells/tissues so it is necessary to include “mouse” in the title.
We modified our title now to ”Study the interaction of effector CD8+ T cells and cortical neurons in response to inflammation in mouse brain slices and neuronal cultures” and have included mouse species.
- Has the viability of neurons in the brain slice culture been accessed?
Concerning the neuronal viability in the brain slice, we did not really quantify. However, we did examine the general viability by checking neuronal morphology in the live cut brain slices of Thy1.2-HcRed mice, in which the intact neurons showed filled cytosolic red fluorescence including clear exons (Supplementary Figure 1). These intact neurons were located in the deeper layer of the slices from the cut surface (15-20 µm deeper). The surface damaged neurons showed no intact red cell signal but fragmented red debris.
- The majority of cells in the neuronal culture are proliferating astrocytes. Have the authors considered to increase the neuron to glia ratio by applying low concentration of cytosine arabinoside?
Thanks for bringing up this essential point. We thought about this at the beginning of the project. However, due to the concern of neuronal damage from the inhibitor treatment which consequently might affect T cell activity, we did not use any inhibitor but allowed neurons to grow with glia cells in the culture. For certain experiments when pure neurons are required, it is indeed necessary to remove excessive astrocytes. Lesslich et al. reported 5-fluoro-2'-deoxyuridine (FUdR) treatment resulted in a higher neuron to astrocyte ratio compared to cytarabino furanoside (AraC) in rat hippocampal cell cultures (Lesslich et al. 2022, DOI: 10.1371/journal.pone.0265084). We definitely will consider using FUdR for future experiments when major neuron population is needed. In this study, we would like to preserve the natural form of neuron-glia mixed culture for T cells as a fundamental experimental template.
- In the figure 2C, was the T cell subpopulation determined with flow cytometry or immunofluorescent staining? Since MOG-specific T cells are isolated from CD8Cre-GFP mice, why are CD8+ T cell subpopulations not first sorted and further co-culture with brain slices?
Data of Fig. 2C was generated from immunostaining showing at the side of the graph. Align with the idea of using mixed neuron-glia culture. We would like to preserve the heterogeneity of effector T cells (explained in line 155-157), as in the autoimmune neuropathology. CD4+ and CD8+ T cells were found at the lesion site of MS patients. Despite using in vitro and ex vivo model, we aimed to create a more native environments to study the interaction of CD8+ T cells and neuronal cells. Therefore, we used transgenic mice to visualize only to target cell population (CD8+ T cells and neurons). Using pure CD8+ T cell population would be indeed critical for certain experiments. However, sorted cells do not functionally perform well based on our own experiences.
- How will be effect on CD8+ T cell migration when adding IFNg or TNFa alone as compared to the T cell condition media?
We have not actually tried IFNg or TNFa alone in our experiments. As we aimed to create models to study T cell-mediated neuropathology, a broad spectrum of cytokine mixed is known to be involved. Therefore, we decided applying single cytokine to the culture would not reconstruct the pathological environment in our study. However, it might be worth to try out if IFNg or TNFa only could induce the observed T cell migration phenotype, or trigger specific inflammatory signaling on certain cell population in detail.
- Figure 4, it might be helpful to fit the data and show the linear or exponential increase.
We calculated the R² values for all the experimental conditions showing in fig.4. Data were fitted with both linear regression and an exponential growth function. An additional table was made to show each R² value (Supplementary Table 1). We embedded the fitting curves in fig.4 with the corresponded text in the result section (line 305-311).
- Line 628, the detailed protocol for imaging analysis (migration parameters) is missing.
We now added the protocol for imaging analysis including the definition of the migration parameters in the methods (line 669-685).
- Discussion, the authors may include some discussions on the potential use of these models to study the change of neural activity/transmission/network under neuroinflammatory conditions.
We added a paragraph in the discussion section accordingly (line 374-386).
Minor points:
- What are the error bars in the figure 2? SEM or SD?
They are mean ± sem. We added the information in the figure legend (line 172).
- The colors showing neurons in the figure 2A (blue) and B (red) are different. It is good to keep consistency.
The Fig. 2A showed actually the mixed neuronal cells with glia cells in a consistency with the data (Fig. 1D). The demonstration was thought to be red neuron, blue astrocytes and yellow microglia.
- On the page 6, there are several places that referred to the wrong figure, e.g., line 207, 227 and 236.
We have corrected them.
- Line 257, “affects” should read as “effects”.
We have corrected it.
- Line 379, what does “NBA medium” stand for? Neural basal?
Yes, it is Neuronal-A medium (now at line 391).
- Line 446, “Microglia” should read as “microglia”.
We have corrected it.

Round 2
Reviewer 2 Report
Thanks for the revised version! Most of my comments have been adequately addressed, however, the following points could be further clarified.
Major points 4:
MOG-specific T cells were isolated and cultured from CD8Cre-GFP mice, so CD8 T cells were supposed to have green fluorescence, but why was CD8-PE antibody used to again label CD8 T cells in the culture? T cells are suspension cells so how was the immunostaining performed (cytospin or Eppendorf tube)? Flow cytometry will be a better way to quantify the percentage of T cells subtypes.
Major points 5:
You may consider to include this point in the discussion.
What is the limitation of these models?
Author Response
We thank again for reviewer’s efforts and valuable suggestions.
Major points 4:
MOG-specific T cells were isolated and cultured from CD8Cre-GFP mice, so CD8 T cells were supposed to have green fluorescence, but why was CD8-PE antibody used to again label CD8 T cells in the culture? T cells are suspension cells so how was the immunostaining performed (cytospin or Eppendorf tube)? Flow cytometry will be a better way to quantify the percentage of T cells subtypes.
Concerning the MOG-specific T cells, we have compared the lymphocyte population generated from WT mice and CD8Cre-GFP mice to ensure the composition under the same protocol is reproducible. The immunostaining data showing in Fig. 2c was MOG T cells generated from WT mice. We apologize for not making the description clear. It is indeed a much better way to quantify cell population by flow cytometry. We performed once by FACS (before our laser disfunctioned) with the MOG T cells isolated from WT mouse. There, the data showed 31% CD4+ T cells and 65% CD8+ T cells, which is comparable to the microscopic results showing in Fig. 2c (25.5±2.0% CD4+ cells and 61.9±2.9% CD8+ cells). The immunostaining results of MOG T cells from the CD8Cre-GFP mice were showing similar population composition as in WT mice in our hands. Due to the disorganized data management, we left the FACS data out from the manuscript. We clarified the preparation and staining protocol in the result (line 148-150) and method section (eppendorf tube; line 563; line 669-675) respectively.
Concerning the CD8-PE labeling in the CD8Cre-GFP cells, in case of Fig. 2c we used WT cells, however, we experienced previously the other commercial CD8Cre-GFP mouse line, in which the GFP is not specifically expressed in CD8+ cell population. Therefore, we always double check if the transgenic labeling is of expected. The CD8Cre-tGFP line we used in the study is very specific. We have checked the GFP+ cells in the spleen, all cells were CD8+ and the immune cell compositions were the same as in WT mice by flow cytometry analysis. Therefore, we expect no difference in the MOG specific cells generation in terms of cell population composition in CD8Cre-tGFP line.
Major points 5:
You may consider to include this point in the discussion.
What is the limitation of these models?
We discussed this point in the discussion (line 388-393).
